# Genome-wide association study of biological nitrogen fixation traits in mini-core cowpea germplasm

Gelase Nkurunziza[1,2*], Emmanuel K. Mbeyagala[3⦿], Emmanuel Amponsah Adjei[1,4,5‡],
Isaac Onziga Dramadri[1,4‡], Richard Edema[1,4‡], Arfang Badji[4‡], Rahiel Hagos Abrah[4,6‡],
Astere Bararyenya[7‡], Kpedetin Ariel Frejus Sodedji[8‡], Phinehas Tukamuhabwa[1‡],
Mildred Ochwo Ssemakula[1‡], John Baptist Tumuhairwe[9⦿], Thomas Lapaka Odong[1⦿]

1 Department of Crop Science and Horticulture, College of Agricultural and Environmental Sciences, Makerere University, Kampala, Uganda, 2 Departement of Research, Institute of Agricultural Sciences of Burundi, Bujumbura, Burundi, 3 National Agricultural Research Organization- National Semi-Arid Resources Research Institute (NARO-NaSARRI), Serere, Uganda, 4 Makerere University Regional Centre for Crop Improvement, Makerere University, Kampala, Uganda, 5 Council for Scientific and Industrial Research - Savanna Agricultural Research Institute, Tamale, Ghana, 6 Department of Plant and Horticultural Sciences, College of Dryland Agriculture and Natural Resources, Mekelle University, Mekelle, Ethiopia, 7 Non-Timber Forest Products and Orphan Crop Species Unit, Laboratory of Applied Ecology (LEA), University of Abomey-Calavi (UAC), Cotonou, Benin, 8 Institute of Advance Training in Agriculture, University of Burundi, Bujumbura, Burundi, 9 Department of Soil Science and Land Use Management, College of Agricultural and Environmental Sciences, Makerere University, Kampala, Uganda

⦿ These authors contributed equally to the work
‡ EAA, IOD, RE, AB, RHA, AB, KAFS, PT and MOS also contributed equally to the work.
* nkurugelase@gmail.com

## Abstract

Biological Nitrogen Fixation (BNF) efficiency in legume crops such as cowpea (*Vigna unguiculata L. Walp*) has been less documented yet is key in improving yield performance and restoring soil fertility in sub-Saharan Africa. Nevertheless, little progress has been made in understanding the gene control of the BNF traits in cowpea to sustain the development of smart agriculture in this part of the world. This study aimed to identify cowpea genotypes and map genomic regions for BNF traits for developing high nitrogen-fixing cultivars. A total of 241 mini-core cowpea genotypes were inoculated with *Bradyrhizobium spp* in a screen house for two cycles. Phenotypic data collected on the number of nodules (NN) per plant, nodule efficiency (NE) in percentage, and nodule dry weight (NDW) per plant revealed significant differences implying high genetic variability in the mini-core population for nodulation capacity. Fifteen significant association signals were identified for BNF traits on nine chromosomes except *Vu02* and *Vu09* when two multi-locus models were considered. Markers accounting for over 15% variation for BNF traits included 2_31410 (2.32Mb) on *Vu05* and 2_45545 (24.93Mb) on *Vu06* for NN, 2_06530 (56.64Mb) and 2_27028 (34.31Mb) on *Vu01* for NE and 2_50837 (10.07Mb) on *Vu01* and 2_11699 (34.41Mb) on *Vu07* for NDW, respectively. Additionally, positional candidate genes near the peak markers

**Data availability statement:** All relevant data are within the paper and its Supporting Information files.

**Funding:** The authors received no specific funding for this work

**Competing interests:** The authors have declared that no competing interest exist.

that encode genes associated with BNF in cowpea included Vigun06g121800, Vigun01g160600, Vigun10g014400, Vigun07g221500, Vigun07g221300 and Vigun11g096700. The genotype TVu-1477 was identified to have favorable alleles for both three studied traits. The significant markers identified in this study can be converted to Kompetitive Allele Specific-PCR (KASP) markers to accelerate the development of high-yielding cowpea varieties that also enhance soil fertility.

## Introduction

The growing need for climate-smart and nutritious crops in the face of increasing soil fertility degradation, exacerbated by climate change in Africa, cannot be overemphasized. Cowpea (*Vigna unguiculata* L. Walp) is a multipurpose legume crops widely grown in sub-Saharan Africa [1]. It is a staple food and a key source of protein for millions of people [2]. Cowpea is well integrated into cropping and rotation systems as a cover crop limiting soil erosion and drought [3,4]. Cowpea is a smart crop with the capacity to fix atmospheric nitrogen through a mutual plant-rhizobium relationship [5–7]. This biological nitrogen fixation (BNF) contributes to its low need for nitrogen fertilizers [8], making cowpea a crucial crop for low-input farming systems [9,10]. However, there is limited understanding of the genetic factors underlying BNF traits in cowpea such as nodule formation, nodule size and weight, and nitrogenase activity. These complex traits are influenced by both environmental and genetic factors [11,12].

The number of nodules (NN) per plant is one of the indicators of BNF in legume crops. It is highly correlated with plant growth [13]. The NN per plant is an indication of BNF ability for different reasons including (i) correlation with symbiotic efficiency as more nodules indicate more sites for conversion of atmospheric nitrogen into forms usable by plants [14], (ii) influence on plant health and growth as legumes with more nodules will show better growth health and vigor resulting in increased grain yield [15] and (iii) soil fertility resulted from increased nitrogen-fixed [16], benefiting the subsequent crops in the rotation systems. However, NN alone is not a clear indicator of BNF efficiency and therefore has to be combined with other BNF-related traits such as function, color, and weight of the nodules [17]. The other BNF indicator is the nodule function or the nodule efficiency which is mainly indicated by their inside color. Active nodules that are functioning in fixing the atmospheric nitrogen show a pink to red color, manifesting the presence of leghaemoglobin, a protein that enhances the transport of oxygen within a living nodule [18]. White, grey or green colors of nodules indicate inefficiency of nodules due to immaturity or senescence of the nodules [14]. BNF efficiency is also indicated by the nodule size which is highly correlated with nodule weight (NDW). The NDW per plant also defined as dry matter content of nodules is the most significant indicator of BNF efficiency as it includes the number and the size of nodules. While nodule fresh weight gives insight into the total biomass of nodules, dry weight correlates effectively with the actual efficiency of BNF in legumes [19].

Over the last four decades, marker-assisted selection (MAS) breeding in plants has been gradually accompanying or replacing the conventional breeding approach [20].

Unlike conventional breeding, the use of DNA markers in MAS breeding offers two advantages including faster recovery of the recurrent genome and a more efficient selection of genomes that have recombination events close to the target gene [20]. By using DNA markers to assist in conventional plant breeding, speed, efficiency and precision are increased [21]. Thus, it has become a routine in many breeding programs to identify meaningful markers that can help achieve higher genetic gain [22,23]. Advances in molecular genetics, particularly Genome-Wide Association Studies (GWAS), offer a powerful tool to dissect the genetic basis [24] of agronomic traits and identify genetic loci that can be leveraged in breeding programs.

Single-nucleotide polymorphisms (SNPs) indicate, in the DNA sequence, a single variation at a nucleotide position, contributing to the function and expression of a gene [25]. On the other hand, GWAS studies scan the entire genome, and identify genetic variants associated with complex traits by comparing SNP frequencies to reveal the gene network associated with those traits [26]. Genomic risk loci, or blocks of associated SNPs that collectively exhibit a statistically significant correlation with the traits of interest, are also commonly reported by GWAS [27].

Various research on GWAS using SNP markers on different traits have been reported in cowpea [28–31]. For instance, more recently, [30] identified 23 SNPs significantly associated with days to flowering (DTF) in cowpea using 51,128 SNPs. They highlighted four key candidate genes: Vigun01g084000, Vigun01g227200, Vigun02g062600, and Vigun03g296800 [30]. Similarly, [29] studied drought tolerance in a Multiparent Advanced Generation Intercross (MAGIC) cowpea population, linking SNPs to plant growth habit, maturity, blooming time, seed weight, and grain yield. Their network-guided method revealed distinct interactions between drought tolerance loci. Likewise, a genome-wide association mapping of 365 cowpea mini-core highlighted three significant regions on chromosomes Vu10, Vu08, and Vu02 and two smaller ones on chromosomes Vu01 and Vu06 to be substantially linked with aphid resistance [28]. The peak SNPs were near Vigun01g233100.1, Vigun02g088900.1, Vigun06g224900.1, Vigun08g030200.1, and Vigun10g031100.1, genes associated with aphid resistance in cowpea. The study provided valuable genetic insights into cowpea flowering time, drought tolerance, and pest resistance. However, information on gene networks associated with BNF traits in cowpea is scanty despite the usefulness of this trait in maximizing cowpea yields [32,33]. Recent progress in dissecting the genetics of BNF using GWAS in legumes was mainly conducted on common bean [12,34] and soybean [35,36]. Identifying candidate genes associated with BNF in cowpea would be essential to enhance sustainable and smart agriculture. Understanding such genes enables cowpea breeders to develop improved cultivars with high nitrogen fixation efficiency, reducing the need for chemical fertilizers, improving soil health, and contributing to food security while addressing environmental concerns.

Therefore, this study aimed to identify cowpea genotypes, genomic regions, and candidate genes associated with BNF traits in the mini-core collection. The cowpea mini-core collected worldwide presents a great potential genetic diversity for investigating marker-trait associations. This research will contribute to a better understanding of the genetics of BNF traits in cowpea-improved cultivars for increased productivity and sustainable agriculture.

## Materials and methods

### Genetic materials and experimental design

This study used 241 genotypes (S1 Table) from the global collection mini-core cowpea population (UCR), maintained by the University of California, Riverside [37] and Makerere University Regional Center for Crop Improvement (MaRCCI). Five varieties released in Uganda, were used as checks. The 241 genotypes were planted in a Randomized Complete Block Design (RCBD) with 2 replications in a screen house at MUARIK in a set of two experiments over time, from November 2022 to January 2023 and November 2023 to January 2024, respectively.

### Soil media preparation

Composite topsoil was sampled from a garden with low nitrogen content (a garden that has not received nitrogen fertilizer or hosted a legume crop for the last 2 seasons). The composite soil was analyzed for pH, organic carbon, electrical conductivity, macronutrients, total nitrogen, extractable phosphorus, and exchangeable bases ($K^+$, $Na^+$, $Mg^{2+}$ and $Ca^{2+}$) using

procedures described by [38]. To harmonize and kill the microorganisms, the soil was steam-sterilized in a drum at 120 °C for six hours and left to cool for 24 hours. To optimize the nodulation efficiency, the soil was then mixed with 975 mg of $KH_2PO_4$, 841.6 mg of KCl, 111.04 mg of $MgSO_4.7H_2O$, 103.12 mg of $ZnSO_4.7H_2O$ and 4.06 mg of $(NH_4)_6 Mo_7 O_{24}.H_2O$ as a source of P, K, Mg, Zn and Mo, respectively [39] for a pot of 5 kg of soil. The soil was then filled in 482 pots with holes at the bottom for aeration and to avoid water logging.

## Inoculum purification, inoculation and planting

The inoculum, *Brady rhizobium spp*, strain ICB756, was purified and incubated in the BNF laboratory at Makerere University following the purification protocol proposed by (Agoyi et al., 2016). We used *Bradyrhizobium spp* which was reported by previous studies [39] as a cowpea-type strain and the most common rhizobial strain found in approximately 70.3% of cowpea-growing soils in Africa [41]. The population grew to 7.91 x $10^9$ cells $g^{-1}$ for experiment 1 and 8.33 x $10^9$ cells $g^{-1}$ for experiment 2. Before planting, seeds of the 241 cowpea mini core genotypes were surface sterilized with 95% alcohol for 10 s to remove waxy substances and air bubbles [42]. To inoculate the seeds, the inoculum was further prepared by dissolving sugar in clean, lukewarm water to create a sticker, which was applied directly to the seeds before planting at a rate of 2x$10^5$ CFU (colony forming unity) per seed. This rate is determined from a mixture of 10 ml of 5% sugar solution mixed with 1 g of rhizobium inoculant made to be mixed with 500 g of cowpea seed [43]. About 5 seeds were sown in each pot, and plants were thinned to two plants per pot after germination. Pots sitting on plastic plates were used to avoid contamination and collect water which was re-poured back once collected.

## Data collection

Data were collected on number of nodules (NN), proportion of efficient nodules (NE) [active nodules] in percentage and nodule dry weight (NDW) per plant at 45–60 days after sowing. This was done by uprooting the plant and carefully separating the root nodules from the plants. The nodules were picked and spread on the sieve to wash and drain water from their surface. The number of nodules (NN) per plant, the percentage of pink to red nodules (NE), and nodule dry weight (dried at 65°C overnight) per plant (NDW) were recorded as recommended by [40,44–46]. The pink-reddish color observed after cutting each nodule reveals the existence of leghemoglobin, a protein-like hemoglobin, which facilitates the transport of oxygen to the nitrogen-fixing bacteria in the nodules (rhizobia) [47].

## Data analysis

**Phenotypic data analysis.** Phenotypic data obtained from the two phenotyping experiments were analyzed using *lmer* function from *lme4* package [48] in R software, version 4.3.2 [49].

Best linear Unbiased Estimators (BLUEs) for all the genotypes for the two screening experiments were obtained using a linear mixed model with genotypes treated as fixed and block as random. The BLUEs were then used in GWAS analysis using the following GWAS model (equation 1):

$$Y = \beta_o + \Sigma\beta_i X_i + \Sigma\phi_j Z_j + \varepsilon \tag{1}$$

Where:

- $Y$ is the phenotypic trait value,

- $\beta_o$ is the intercept of the model

- $\beta_i$ is the effect size (regression coefficient) of the $i^{th}$ SNP genotype (coded as 0,1 or 2 for the number of minor alleles),

- $X_i$ is the genotype score for $i^{th}$ SNP,

- $\phi_j$ is the effect size of the j[th] covariate

- $Z_j$ is the value of the j[th] covariate for each individual (population structure),

- ε is the error term associated with all other unmeasured factors influencing the trait

Phenotypic variation, genotypic variation, broad sense heritability (genetic variance/phenotypic variance), and predicted genetic advance were estimated using the variance components for BNF traits. Furthermore, frequency distributions based on the nodulation capacity of the genotypes under test were produced by the analysis and used to characterize the phenotypic distribution of the genotypes.

**Genotyping of the mini-core population.** This min-core cowpea population has been genotyped at the University of Southern California, Molecular Genome Core facility (Los Angeles, CA, USA) using the Cowpea iSelect Consortium Array containing 51,128 SNPs [37]. SNPs were called using Illumina, Inc.'s GenomeStudio software V.2011.1 (San Diego, CA, USA), and their precise locations were ascertained using the IT97K-499–35 reference genome v1.0 [50,51]. Using TASSEL version 5.2.94 [52], SNP data were filtered for downstream analysis, removing SNPs of more than 20% missing data and minor allele frequency (MAF) of less than 5%, resulting in a total of 40,246 SNPs. Thereafter, a SNP density map was created using the memory-efficient, visualization-enhanced, parallel-accelerated R package "rMVP" [53], with a window size of 1Mb, to investigate the distributions of the 40,246 SNPs on the 11 chromosomes of cowpea.

**Marker trait association analysis.** Marker trait association analysis was performed in mrMLM.GUI (Multi-Locus Random-SNP-Effect Mixed Linear Model Tools for Genome-Wide Association Study with Graphical User Interface) package, version 4.0.2 [54]. Six different GWAS models were tested including (i) multi-locus mixed linear model -mrMLM- [55], (ii) FAST multi-locus random SNP effect mixed linear model FASTmrMLM [56], (iii) FASTmrEMMA: FAST multi-locus random effect mixed model analysis [57], (iv) pLARmEB: polygenic linkage disequilibrium adjusted random mixed linear model with empirical Bayes [58], (v) pKWmEB: polygenic Kruskal-Wallis test with empirical Bayes [59] and (vi) ISIS EM-BLASSO: Iterative Sure Independence Screening EM Bayesian LASSO [60]. The Manhattan and the quantile-quantile (QQ) plots were generated using the same mrMLM.GUI package. The advantage of mrMLM.GUI package is that the genome is examined for all markers, the significance threshold is not as strict while the false positive rates are well-controlled [57]. Hence, the choice of the mrMLM.GUI package with its six integrated GWAS models was based on its efficiency in detecting loci with small effects in multi-locus GWAS. Thus, mrMLM. GUI is better than other multi-locus models such GAPIT, TASSEL and FamCPU due to its (i) multi-locus detection, (ii) GUI-based user interface, (iii) very high power for polygenic traits, (iv) low false positive rate and (v) advanced environmental covariate [54,61]. Then, a multi-locus genetic model is constructed using all the markers that may be associated with the trait, the effects of each marker being assessed using empirical Bayes, and any nonzero effects being further identified using the likelihood ratio test for true QTL [57]. The Bonferroni correction factor, obtained by considering a significance level of 0.05 (0.05/number of markers), was applied to the p-value of each tested SNP to detect highly significant Marker-Trait Associations (MTAs). Also, since multi-locus GWAS approaches do not involve multiple testing, a logarithm of odds (LOD) score threshold of 3.0 was considered to declare additional significant MTAs. Additionally, significant markers picked by at least two GWAS models were reported in this study.

**Identification of candidate genes.** Using the genome browser (JBrowse) in Phytozome 13 (https://phytozome-next.jgi.doe.gov/info/Vunguiculata_v1_2, accessed on 23rd August 2024) the positions of peak SNPs were searched along the annotated genome (v1.2) of elite IITA cowpea variety IT97K-499–35 [50] to investigate the likely genes responsible for the detected association signals. Genes were considered candidates if they were within ± 80 kb region from significant SNP, with the LD decay rate for mini-core population [37] and performed a function related to BNF. A further investigation of the predicted genes found in the peak SNP regions was conducted to determine their annotated biological function compared with homologs in other legumes such as common bean (*Phaseolus vulgaris*) and soybean (*Glycine max*) as well as Arabidopsis thaliana. To ascertain the roles of the genes associated with the various SNPs found, the European Molecular Biology Laboratory European Bioinformatics Institute (EMBL-EBI) public database InterPro was used [62]. Using the "*ggplot2*" package, version 3.5.1, the SNPs' contributions to BNF traits based on the observed alleles were plotted, and the package "*rstatix*", version 0.7.2, was used to compute their confidence statistics.

## Results

### Phenotypic variation assessment

The cowpea mini core genotypes evaluated for BNF traits showed significant variability (p<0.001) for both NN, NE and NDW across the two screening experiments conducted (Table 1). For all three traits studied, the mean performances were 27 nodules per plant, 50.11%, and 37.5 mg/plant for NN, NE and NDW, respectively higher in experiment one than in experiment two where mean performance of 22.6 nodules per plant, 38.7% and 26.2 mg/plant, respectively were recorded for NN, NE and NDW. The genotypes exhibited high significant differences (p<0.001) throughout the experiments for all the studied BNF traits. Similarly, the broad sense heritability estimates ranging from 0.76% to 0.84% were recorded in experiment one higher the range of 0.54% to 0.62% recorded in experiment two and the range of 0.25% to 0.28% for the combined analysis of all BNF traits (Table 1).

For all traits in both experiments, most BNF-efficient genotypes originated from Africa, followed by those from Asia and America (Fig 1). Contrarily, genotypes from Australia and Europe were less efficient for BNF traits, the Australian ones being the worst (Fig 1).

### Marker coverage and distribution

The 40,246 polymorphic SNP markers are distributed on the 11 chromosomes of cowpea according to the distribution shown in Fig 2. The total number of SNPs per chromosome ranged from 2,911 SNPs on chromosome 2–6,270 SNPs on chromosome 3.

### Genetic variation assessment

To assess the genetic diversity and structure within the 40,246 polymorphic SNP markers, a phylogenetic tree was constructed using the Neighbor-Joining (NJ) method based on the kinship matrix derived from the SNP dataset. The phylogenetic tree shows that the 241 cowpea genotypes are divided into 5 clusters (Fig 3A). Genotypes grouped under Cluster I were the most predominant (40.2%), followed by those under Cluster II (22.8%), unlike Cluster V, which had few accessions (8.3%). Similarly, genotypes grouped in Cluster I were genetically related to the genotypes clustered in Cluster II rather than to those in Cluster III and IV (Fig 3A & B). On the other hand, PC1 (17.37%) explained most of the genetic variance, followed by PC2 (6.6%) and PC3 (4.09%) for the 40,246 polymorphic SNP markers evaluated (Fig 3B).

### Marker trait association analysis

In this study, 15 significant association signals were detected for BNF traits including NN, NE and NDW in the mini-core cowpea accessions using at least two multi-locus GWAS models.

*Combined analysis.* In the combined analysis, five significant GWAS signals were detected on chromosomes Vu01, Vu10 for NE and Vu03 and Vu07 for NDW. For NE, SNP marker 2_29403 (34.26Mb) on Vu01, with MAF 0.47 (LOD = 5.8) was depicted by mrMLM, FASTmrMLM, pLARmEB and ISIS EM-BLASSO models with a QTN effect ranging from 9.1 to 10.3, explaining a total phenotypic variation ranging from 23.9 to 30.6%. SNP marker 2_06530 (56.54Mb) on Vu03 with MAF of 0.35 (LOD = 3) was depicted by mrMLM, FASTmrMLM and pLARmEB models and had a QTN effect ranging from -7.2 to -5.6 explaining a total variation ranging from 8.4 to 13.4%. On the other hand, SNP marker 2_20695 (1.59Mb) on Vu10 with MAF of 0.06 (LOD= 3.2) was depicted by mrMLM, FASTmrMLM and pLARmEB models with a QTN effect ranging from 11.3 to 15.2, explaining a total phenotypic variation ranging from 7.9 to 14.2% (Table 2, Fig 4). The GWAS scan for NE identified three alleles including GG, TT, and AA (Table 2. Fig 4).

In the case of NDW, marker 2_50837 (10.07Mb) on Vu03 with MAF of 0.21 (LOD= 3.2) was depicted by mrMLM and FASTmrMLM models with a QTN effect ranging from -20.8 to 18.1, explaining a total variation ranging from 37.7 to 46.2%. The SNP marker 2_20053 (34.43Mb) on Vu07 with MAF of 0.03 (LOD=7.6) was identified by mrMLM, pLARmEB and ISIS

**Table 1. Analysis of variance and summary of the descriptive statistics for the BNF-related studied traits of 241 cowpea genotypes.**

**Experiment one**

| Source | Df | Mean square | | |
|---|---|---|---|---|
| | | NN | NE | NDW |
| Rep | 1 | 166.8 | 488 | 2238 |
| Genotype | 240 | 951.4*** | 1590.6*** | 2861.7*** |
| Residuals | 240 | 151 | 377.4 | 670.2 |
| Mean ±SE | | 27.04±8.68 | 50.11±13.74 | 37.55±18.3 |
| CV (%) | | 45.43 | 38.75 | 70.82 |
| LSD (5%) | | 24.2 | 38.26 | 50.99 |
| $H^2$ | | 0.84 | 0.76 | 0.76 |

**Experiment two**

| Source | Df | Mean square | | |
|---|---|---|---|---|
| | | NN | NE | NDW |
| Rep | 1 | 1455.3 | 265.2 | 333 |
| Genotype | 240 | 708.3*** | 1144.9*** | 1265.5*** |
| Residuals | 240 | 266.7 | 518.7 | 514.5 |
| Mean ±SE | | 22.57±11.55 | 38.76±16.1 | 26.23±16.03 |
| CV (%) | | 72.33 | 58.78 | 86.47 |
| LSD (5%) | | 32.16 | 44.86 | 44.68 |
| $H^2$ | | 0.62 | 0.54 | 0.59 |

**Combined**

| Source | Df | Mean square | | |
|---|---|---|---|---|
| | | NN | NE | NDW |
| Experiment | 1 | 4816.55*** | 31186.68*** | 25656.44*** |
| Genotype | 240 | 968.99*** | 1568.09*** | 2374.30*** |
| Gen*Exp | 240 | 690.78*** | 1167.38*** | 1752.85*** |
| Residuals | 480 | 208.84 | 448.04 | 592.36 |
| Mean ±SE | | 24.8±13.16 | 44.43±17.12 | 31.4±20.97 |
| CV (%) | | 58.25 | 47.64 | 77.53 |
| $H^2$ | | 0.28 | 0.25 | 0.26 |

NN: number of nodules per plant, NE: percentage of nodule efficiency, NDW: Nodule dry weight in mg/plant, MS: Mean square, SE: standard error of genotype mean, CV: Coefficient of variation, $LSD_{\alpha=0.05}$: least significant difference at 5% confidence level, $H^2$: Broad sense heritability, ***: significantly different at p<0.001.

EM-BLASSO models with a QTN effect ranging from -27.0 to -9.3, explaining a total phenotypic variation ranging from 7.4 to 26.3% (Table 2, Fig 4).

**Experiment two.** In experiment two, the multi-locus GWAS models identified six association signals, five for NN and one for NDW. SNP marker 2_45545 (24.93Mb) on *Vu06* with MAF of 0.09 (LOD=5.4) was identified using mrMLM and FASTmrMLM models with a QTN effect ranging from -19.6 to -13.0 and explained a total phenotypic variance from 18.4% to 41.5%. Two markers were identified on *Vu03*. The first one includes 2_20073 (7.51Mb) with MAF of 0.35 (LOD =4.1) was identified by FASTmrEMMA, pLARmEB and pKWmEB models, with a QTN effect ranging from -14.4 to -6.6, explaining a total phenotypic variance ranging from 9.9 to 16.1%. The second marker, 2_00537 (39.36Mb) with MAF of 0.11 (LOD= 4.7) was identified using ISIS EM-BLASSO and pLARmEB models, with a QTN effect ranging from -8.5 to -7.2, explaining a total phenotypic variation of 5.5 to 7.7%. Marker 2_26983 (5.73Mb) with MAF of 0.19 (LOD=7.2) was identified using ISIS EM-BLASSO and pLARmEB models with QTN effects ranging from -8.9 to -7.1, explaining a total phenotypic variance of 11.0

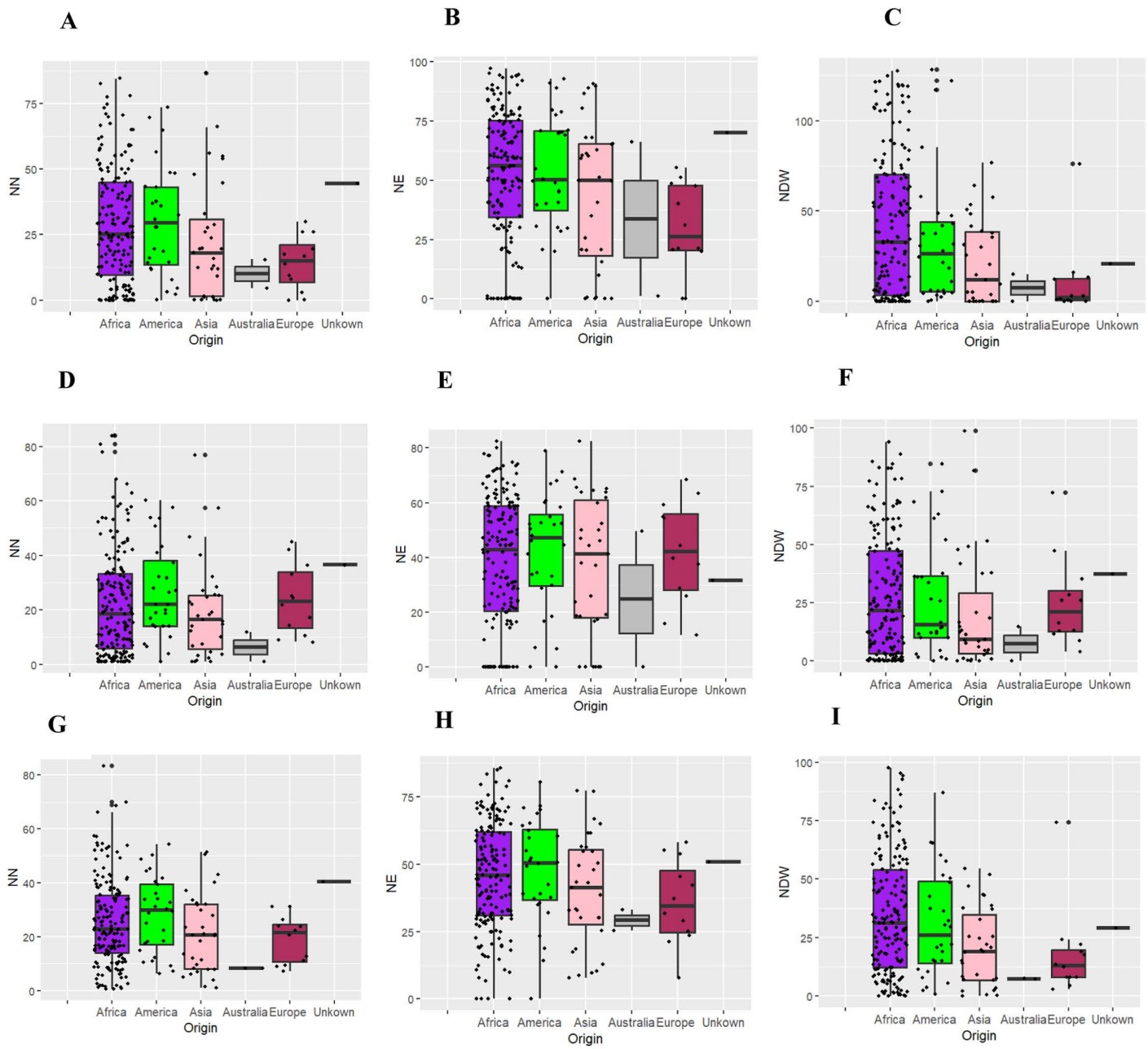

**Fig 1. Trait distribution by origin of 241 cowpea genotypes studied based on NN (A, D&G), NE (B, E&H) and NDW (C, F&I) in experiments 1, 2 and combined, respectively.** Dots represent the cowpea genotypes evaluated.

to 15.2%. Marker 2_52781 (27.31Mb) on *Vu04* with MAF of 0.17 (LOD=6.0) was detected by FASTmrEMMA and pKWmEB models with a QTN effect ranging from 6.9 to 25.6, explaining a total phenotypic variance of 9.3 to 32.1%. Marker 2_26983 (5.73Mb) on *Vu05* with MAF of 0.19 (LOD= 7.2) was depicted by ISIS EM-BLASSO and pKWmEB models with a QTN effect ranging -8.9 to -7.1, explaining a total phenotypic variance of 11.0 to 15.2% (Table 2, Fig 5).

For NDW, marker 2_39541 (28.19Mb) was identified on *Vu11* with MAF of 0.25 (LOD=5.4) using four models including FASTmrMLM, FASTmrEMMA, ISIS EM-BLASSO, and pLARmEB models with a QTN marker effect ranging from -33.5 to -15.5 and explained a total phenotypic variance ranging from 26 to 37.7% (Table 2, Fig 5).

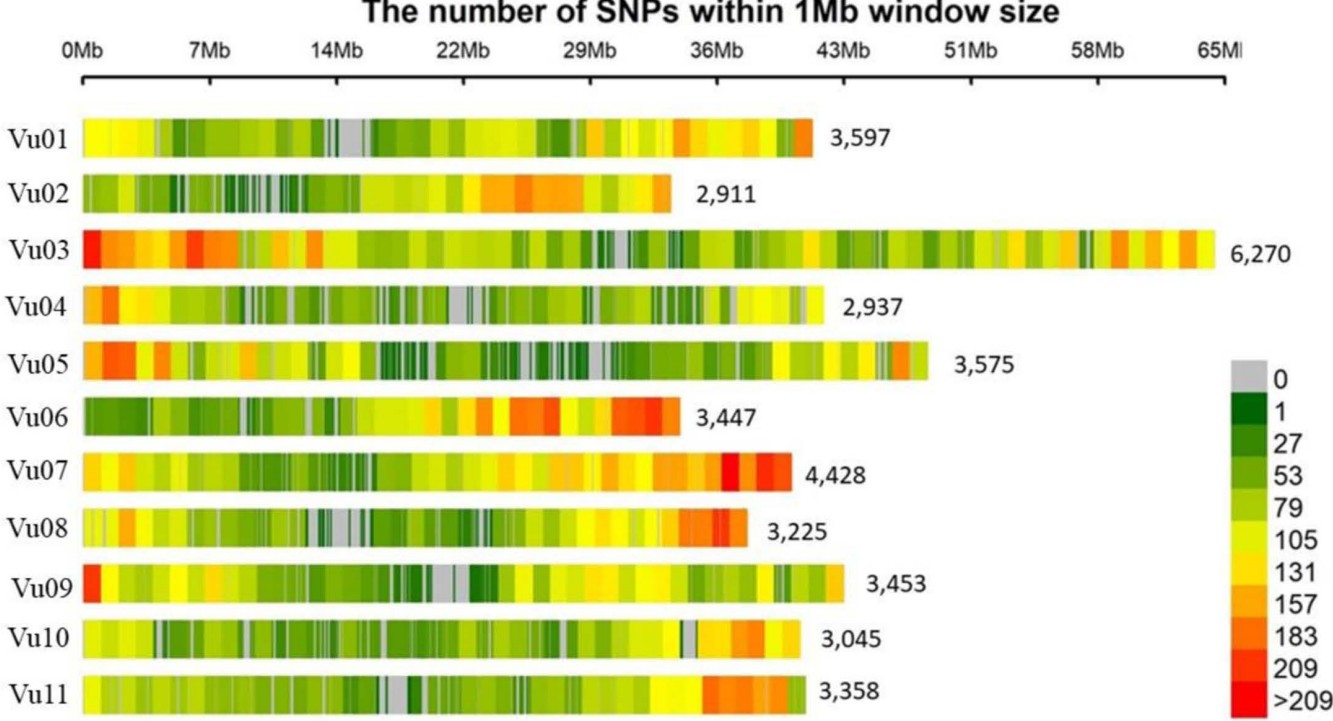

**Fig 2. Distribution of SNPs on 11 chromosomes of cowpea within 1Mb window size.** The horizontal axis shows chromosome length with their respective SNPs density. The legend (0-209) indicates the SNPs density.

**Experiment one.** In experiment one, the multi-locus GWAS models used detected four significant association signals for NN, NE and NDW. For NN, marker 2_31410 (2.32 Mb) on *Vu05* with MAF of 0.42 (LOD= 4.2) was identified using two models including FASTmrEMMA and pKWmEB models with a QTN effect ranging from 11.5 to 24.1 and explained total phenotypic variation ranging from 26.6 to 29.3% (Table 2, Fig 6). Marker 2_27028 (34.31 Mb) was detected on *Vu01* for NE, with MAF of 0.33 (LOD= 3.5) was depicted by FASTmrMLM and ISIS EM-BLASSO models with a QTN effect ranging from 12.5 to 13.0 and explaining a total phenotypic variation of 22.9 to 24.9% (Table 2, Fig 6). Two ML GWAS signals were identified for NDW on chromosomes *Vu07* and *Vu08*. SNP marker 2_11699 (34.41 Mb) on *Vu07* with MAF of 0.18 was (LOD= 8.2) was identified by five models including mrMLM, FASTmrMLM, FASTmrEMMA, ISIS EM-BLASSO and pKWmEB models with a QTN effect ranging from -65.7 to -15.9 and explaining a total phenotypic variation ranging from 31.7 to 48.2% (Table 2). Marker 2_11831 (36.58 Mb) on *Vu08* with MAF of 0.25 (LOD =4.1) was depicted by FASTmrMLM, ISIS EM-BLASSO and pKWmEB models with a QTN effect ranging from -21.3 to -16.8 and explaining a total phenotypic variation ranging from 12.0 to 15.1% (Table 2, Fig 6).

For NDW, the Quantile-Quantile (QQ) plot demonstrated a moderate deviation from the expected distribution of the observed p-value, confirming a potential association of SNPs with the NDW in the population (Fig 6).

### Gene prediction and function (annotation)

Among the significant SNPs identified, only six were found to be associated with meaningful biological functions related to BNF. For NN, one candidate gene was identified on chromosome 6. The identified gene, Vigun06g121800 (24.90 Mb), is 26kb upstream of marker 2_45545 (24.93 Mb) and encodes for Auxin-responsive GH3 family protein (Table 3).

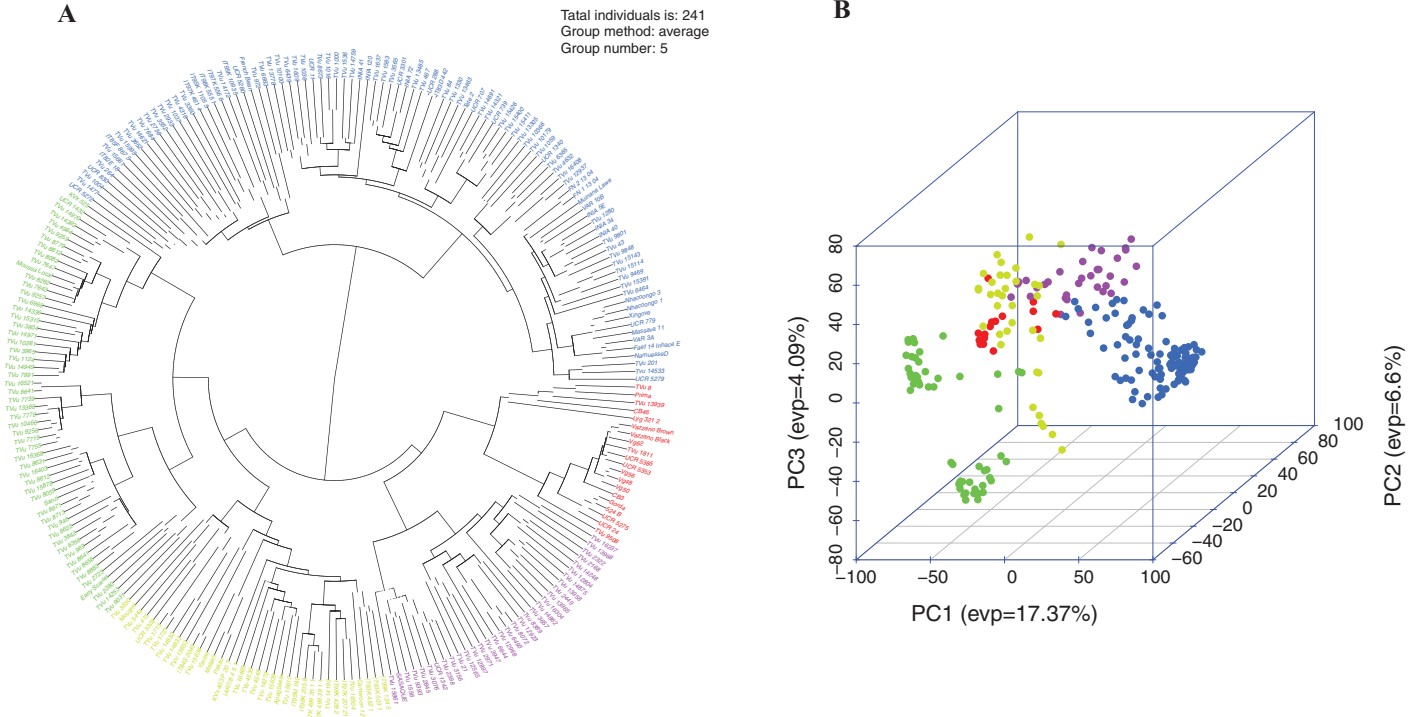

**Fig 3. Neighbor-Joining (NJ) phylogenetic tree based on the kinship matrix derived from the genotypic data (A) and three-dimensional distribution depicted by principal component (PC) analysis (B) of the 241 cowpea genotypes.**

For NE, two candidate genes were reported; Vigun01g160600 was on chromosome 01 at a position near Vu_34265294 and Vigun10g014400 located on chromosome 10 at a position nearing Vu01_1598130 (Table 3).

For NDW, three genes were found, one near the Vu07_34414059 (Vigun07g221500) on chromosome 7, the second one was on chromosome 7 at a position near Vu07_34437020 (Vigun07g221300) and the third one was on chromosome 11, near Vu11_28197064 (Vigun11g096700).

The gene Vigun10g014400, located at V10_1598130 and the gene Vigun07g221500, located at Vu7_34414059 have the same biological function related to BNF (Table 3).

A summary of their respective role is presented in Table 3.

## Marker effect on BNF studied traits

The identified positions with a biological function related to BNF were further dissected to investigate their homozygosity or heterozygosity status. Thus, cowpea genotypes with the homozygous allele AA (Vu06_24933702: snp2_45545) on chromosome 6 had more NN (Fig 7A) than those of GG with a non-allelic significant difference (p = 0.099). In terms of NE, genotypes with homozygous allele GG on chromosome 1 (Vu01_34265294: snp2_29403) were more efficient than those with homozygous allele AA (Fig 7B) and on chromosome 10 (Vu10_1598130: snp2_20695), genotypes with homozygous allele TT are more efficient than those of GG (Fig 7C) with a non-allelic significant difference (p = 0.34 and p = 0.63, respectively). In the case of NDW, genotypes with homozygous allele GG on Vu07_34414059: snp2_11699 had a higher nodule dry weight per plant than those of AA allele (Fig 7D) with an allelic significant difference (p<0.001) while on the same chromosome 7, in position Vu07_34437020: snp2_20053, the marker effect revealed that genotypes with

**Table 2. Trait, SNP ID, allele, chromosome, position, model, QTN effect, LOD score, proportion of phenotypic variation explained (r²), and minor allele frequency of the most significant single nucleotide polymorphisms (SNPs) for BNF-related traits measured on cowpea mini core population evaluated in the screen house at MUARIK, Uganda.**

| Experiment | Trait | SNP ID | Allele | Chr | Marker Position (bp) | Model | QTN effect | LOD score | -log10(P) | r2 (%) | MAF |
|---|---|---|---|---|---|---|---|---|---|---|---|
| Combined[a] | NE | 2_29403 | G/A | 1 | 34265294 | 1,2,4,6 | 9.1-10.3 | 5.8 | 6.7 | 23.9-30.6 | 0.47 |
| | NE | 2_06530 | G/A | 3 | 56,549,146 | 1,2,4 | (-7.2) - (-5.6) | 3 | 3.7 | 8.4-13.4 | 0.35 |
| | NE | 2_20695 | TT | 10 | 1,598,130 | 1,2,4 | 11.3 - 15.2 | 3.2 | 3.9 | 7.9-14.2 | 0.06 |
| | NDW | 2_50837 | C/T | 3 | 10,079,842 | 1,2 | (-20.8) - (-18.1) | 3.2 | 3.9 | 37.7-46.2 | 0.21 |
| | NDW | 2_20053 | T/C | 7 | 34,437,020 | 1,4,6 | (-27.0) - (-9.3) | 7.6 | 8.4 | 7.4-26.3 | 0.03 |
| Exp2[b] | NN | 2_45545 | A/G | 6 | 24,933,702 | 1,2 | (-19.6) - (-13.0) | 5.4 | 6.2 | 18.4-41.5 | 0.09 |
| | NN | 2_20073 | G/T | 3 | 7,510,199 | 3,4,6 | (-14.4) - (-6.6) | 4.1 | 4.8 | 9.9-16.1 | 0.35 |
| | NN | 2_52781 | G/A | 4 | 27,315,973 | 3,6 | 6.9 - 25.6 | 6 | 6.8 | 9.3-32.1 | 0.17 |
| | NN | 2_00537 | G/A | 3 | 39,364,008 | 4,6 | (-8.5) - (-7.2) | 4.7 | 5.5 | 7.7 | 0.11 |
| | NN | 2_26983 | A/C | 5 | 5,734,744 | 4,6 | (-8.9) - (-7.1) | 7.2 | 8.1 | 11.0-15.2 | 0.19 |
| | NDW | 2_39541 | G/A | 11 | 28,197,064 | 2,3,4,6 | (-33.5) - (-15.5) | 5.4 | 6.2 | 0.26-37.7 | 0.25 |
| Exp1[c] | NN | 2_31410 | C/A | 5 | 2,329,672 | 3,4 | 11.5 - 24.1 | 4.2 | 5 | 26.6-29.3 | 0.42 |
| | NE | 2_27028 | G/T | 1 | 34,311,782 | 2,4 | 12.5 - 13.0 | 3.5 | 4.2 | 22.9-24.9 | 0.33 |
| | NDW | 2_11699 | G/A | 7 | 34,414,059 | 1,2,3,4,6 | (-65.7) - (-15.9) | 8.2 | 9.1 | 31.7-48.2 | 0.18 |
| | NDW | 2_11831 | T/C | 8 | 36,587,185 | 2,4,6 | (-21.3) - (-16.8) | 4.1 | 4.8 | 12.0-15.1 | 0.25 |

**Exp1**, experiment one; **Exp2**, experiment two; **NN**, number of nodules per plant; **NE**, percentage of nodule efficiency; **NDW,** Nodule dry weight in mg/plant. Model: **(1),** mrMLM; **(2),** FASTmrMLM; **(3),** FASTmrEMMA; **(4),** ISIS EM-BLASSO; **(5),** pLARmEB; **(6),** pKWmEB; **CHR,** Chromosome; **bp,** base pair; **QTN,** quantitative trait nucleotides; **LOD,** Logarithm of odds; **r2,** Percentage variation explained by the QTN; **MAF,** Minor allele frequency;

[a]: Full table of SNPs identified in combined analysis is presented in S2 Table;

[b]: Full table of SNPs identified in Experiment two is presented in S3 Table;

[c]: Full table of SNPs identified in Experiment one is presented in S4 Table.

homozygous allele TT had high NN than those of CC (Fig 7F) with a non-allelic significant difference (p=0.7). However, on chromosome 11, in position Vu11_28197064: snp2_39541, genotypes with high NDW were with homozygous alleles of AA and GG (Fig 7E) with an allelic significant difference (p<0.01).

Genotypes such as INIA-40, TVu-14971, TVu-1477, TVu-10366, KVx-525, TVu-3965 had both favorable alleles for NN and NDW, genotypes such as TVu-14691 and TVu-1477 had favorable for NN and NE, genotypes such as 524-B, TVu-16408, TVu-14321 and TVu-1477 had favorable alleles for NE and NDW while the only genotype TVu-1477 had favorable allele for both NN, NE and NDW (Table 4).

## Discussion

Breeding for biological nitrogen fixation (BNF) efficiency in legume crops requires understanding the variation of BNF indicators in a given population [70]. Those indicators include the number of nodules per plant (NN), nodule efficiency (NE) in percentage, and nodule dry weight per plant (NDW) among others. In the present study, the phenotypic variation assessment of the 241 genotypes revealed significant variations for the three nodulation traits studied suggesting that the mini-core germplasm panel is well suited for exploring the genetic control of those key traits [28]. On the other hand, the low to moderate heritability (0.28–0.84) for nodulation recorded in this study is comparable to those reported in soybean (0.39–0.98) by [71,72] and common bean (0.25–071) [73,74]. Such moderate heritability implies, however, that, in addition to genetic factors, environmental conditions and specific interactions with rhizobia strain also play a non-negligible role in influencing nodulation [75], suggesting that cowpea breeders should consider genetic and environmental aspects when improving nodulation efficiency.

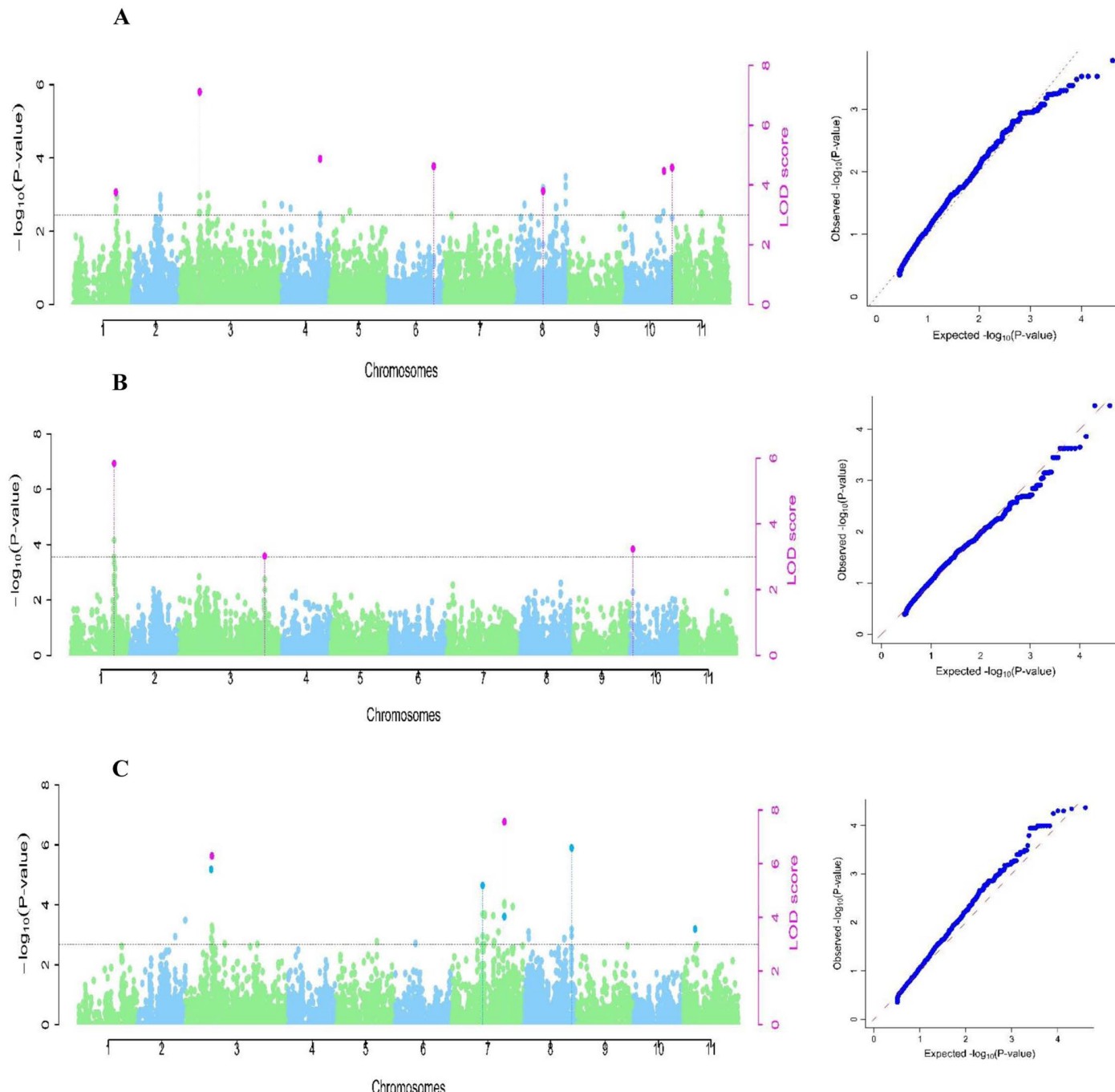

**Fig 4. Manhattan plots for combined analysis and respective quantile-quantile (QQ) plots for the cowpea mini-core population.** The x-axis shows the 11 cowpea chromosomes with physical positions, the y-axis displays the −log10 (p)-values. The threshold with a critical logarithm odd of 3. A= number of nodules per plant (NN); B= percentage of nodule efficiency (NE); C= Nodule dry weight in mg/plant (NDW).

For NN, 6 (1 in Exp1 and 5 in Exp2) significant SNPs were found, but most of them were not associated with the nodulation process in legumes. Only one genomic position on chromosome 6 (Vu06) was associated with nodulation activities. The associated gene, Vigun06g121800 (GH3) is specifically responsible for regulating the auxin hormones, essential for

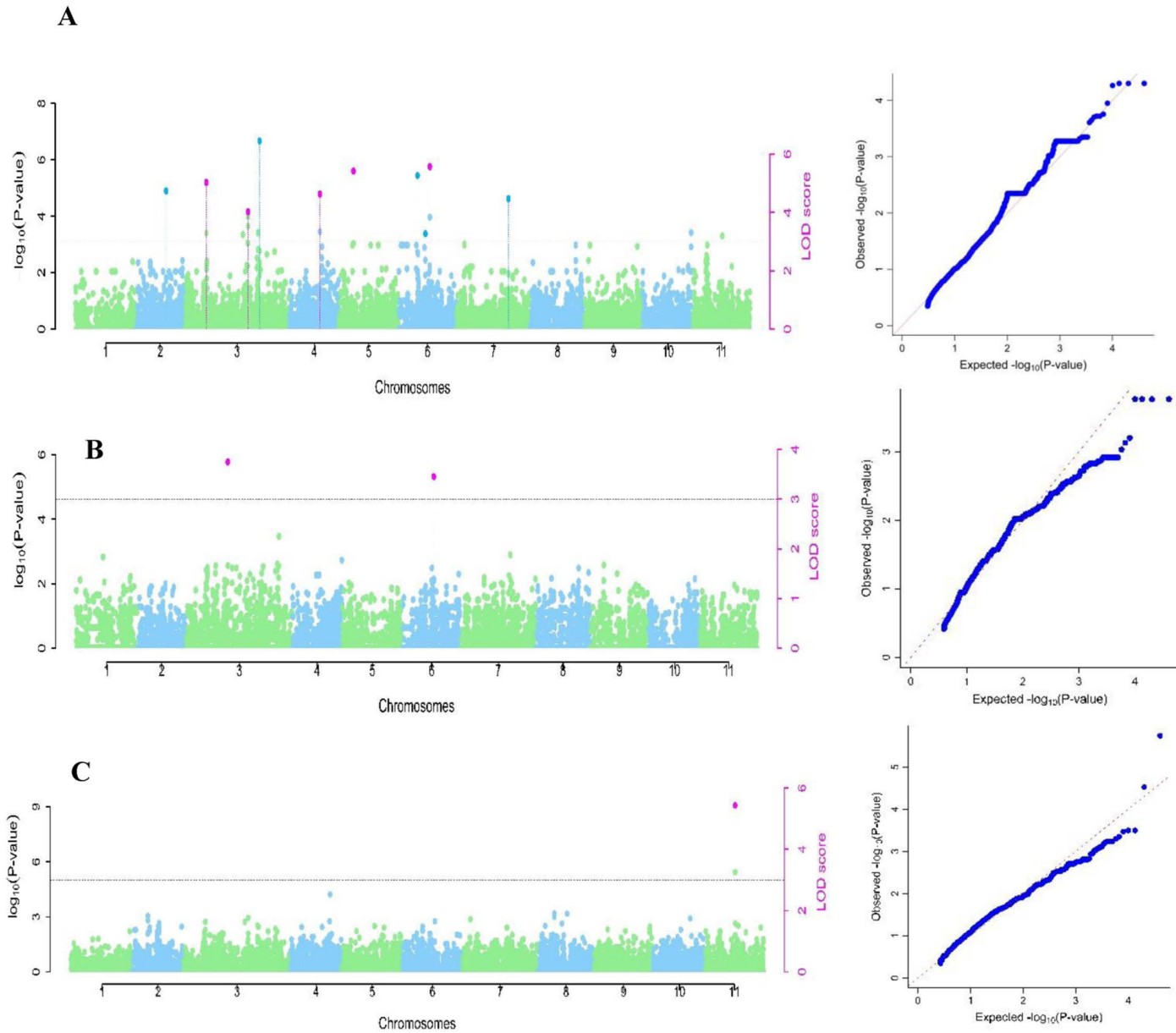

**Fig 5. Manhattan plots for experiment two and respective quantile-quantile (QQ) plots for the cowpea mini-core population.** The x-axis shows the 11 cowpea chromosomes with physical positions, the y-axis displays the −log10 (p)-values. The threshold with a critical logarithm odd of 3. A= number of nodules per plant (NN); B= percentage of nodule efficiency (NE); C= Nodule dry weight in mg/plant (NDW).

nodule formation and BNF process [63,64]. In the soybean genome, members of the GH3 gene family are distributed across several chromosomes, including adjacent genes such as *GmGH3–7, -8. -9 and -10*, located on chromosome 6 [76], corroborating our findings. In our investigation, this gene was located at Vu06_24911984 (21.7 kb) downstream away from the SNP marker flagged by Vu06_24933702.

For NE, two genomic regions (on chromosomes Vu01 and Vu10) were detected. On Vu01, the identified gene was *Vigun01g160600* located at 8.3kb downstream of the actual position (Vu01_34265294) of the signal scanned by GWAS. That position is linked with the PfkB gene (phosphofructokinase B) which is essential in maintaining the energy

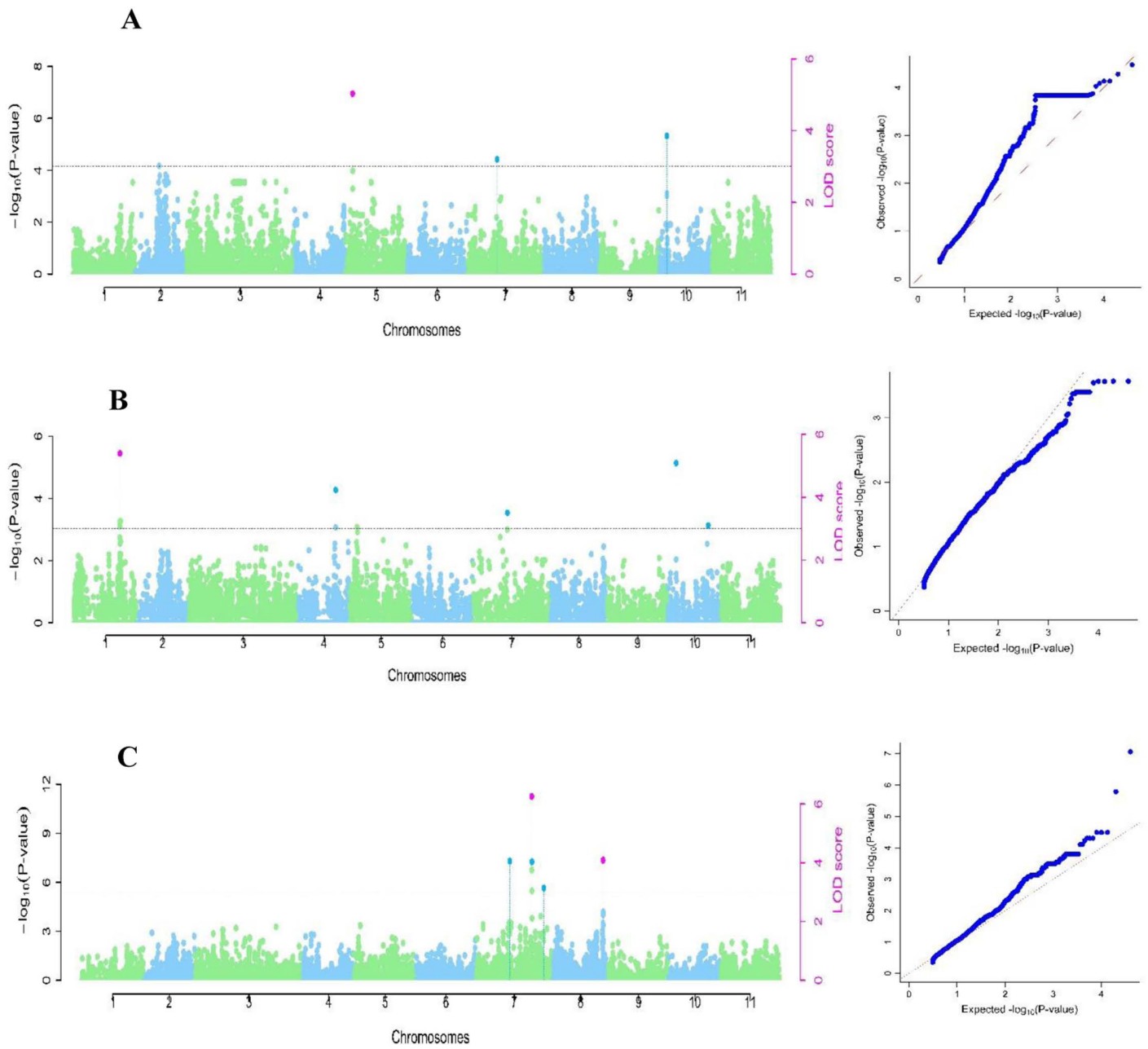

**Fig 6. Manhattan plots for experiment one and respective quantile-quantile (QQ) plots for the cowpea mini-core population.** The x-axis shows the 11 cowpea chromosomes with physical positions, the y-axis displays the −log10 (p)-values. The threshold with a critical logarithm odd of 3. A= number of nodules per plant (NN); B= percentage of nodule efficiency (NE); C= Nodule dry weight in mg/plant (NDW).

supply for root nodule activity and therefore plays a great role in nodulation efficiency [65]. In soybean, the same PfkB gene has been reported as a carbohydrate kinase encoding gene (*Glyma02g245800*) on chromosome 2, and supports energy supply and carbohydrate metabolism, essential for nodule activity, enhancing nitrogen fixation efficiency [77]. The second significant genomic region associated with nodule efficiency was identified on chromosome Vu10 at Vu10_1598130, 9.05 kb upstream of the actual GWAS-scanned position. The associated genes, *Vigun10g014400.1* and

**Table 3. Gene annotation for the significant identified SNPs for BNF traits of the mini-core cowpea population.**

| Trait | Chr | SNP Position (bp) | Gene ID | SNP marker ID | Function * | Reference |
|-------|-----|-------------------|---------|---------------|------------|-----------|
| NN | Vu06 | 24,933,702 | Vigun06g121800 | 2_45545 | GH3 genes regulate the auxin hormones, essential for nodule formation and BNF process | [63,64] |
| NE | Vu01 | 34,265,294 | Vigun01g160600 | 2_29403 | The PfkB is essential in maintaining the energy supply for root nodule activity | [65] |
| | Vu10 | 1,598,130 | Vigun10g014400 | 2_20695 | The wide family of transcription factors known as the WRKY gene family is crucial to the growth and development | [66] |
| NDW | Vu07 | 34,414,059 | Vigun07g221500 | 2_11699 | The wide family of transcription factors known as the WRKY gene family is crucial to the growth and development | [66] |
| | Vu07 | 34,437,020 | Vigun07g221300 | 2_20053 | SPX gene is involved in developmental processes such as root development | [67] |
| | Vu11 | 28,197,064 | Vigun11g096700 | 2_39541 | Thoumatin regulates the specificity of rhizobial strains in symbiotic nitrogen fixation and intervenes in the developmental process | [68,69] |

NN: number of nodules per plant, NE: percentage of nodule efficiency, NDW: Nodule dry weight in mg/plant, *: Full table of gene information in cowpea on the selected regions is presented in S5 Table.

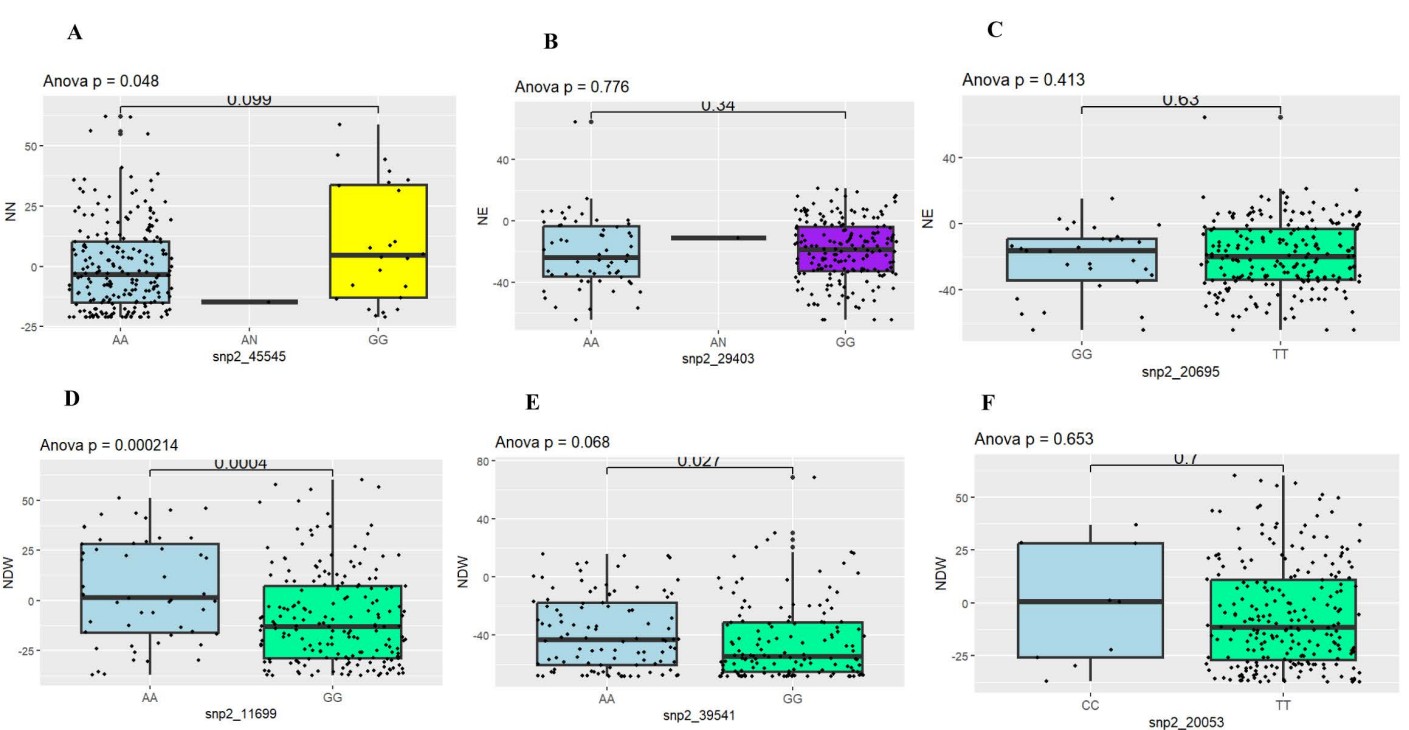

**Fig 7. Boxplots showing the effect of the selected significant markers with a biological function related to BNF-related traits (NN, NE and NDW) on chromosome 1** (B), chromosome 6 (A), chromosome 7 (D&F), chromosome 10 (C) and chromosome 11 (E). The x-axis represents the allele variants (A, C, G and T).

*Vigun10g014400*.2, belong to the WRKY gene family, which is involved in plant growth, development and stress response across various species [66,78–80]. Likewise, one of the three positions identified for nodule dry weight per plant (NDW), Vu7_34414059, was associated with this same gene family. The genes associated were identified as *Vigun07g221500.1* and *Vigun07g221500.2*, located 0.6 kb upstream of the actual GWAS-scanned position. Given their functional similarity to

**Table 4. Profile of the alleles at the top 15 selected (for NN, NE, and NDW) genotypes from the 241cowpea genotypes evaluated.**

| Gen | NN | Gen | NE | | Gen | NDW | | |
|---|---|---|---|---|---|---|---|---|
| | 2_45545 | | 2_29403 | 2_20695 | | 2_11699 | 2_20053 | 2_39541 |
| **INIA-40** | AA | **524-B** | AA | TT | **524-B** | GG | TT | GG |
| **TVu-14971** | AA | **TVu-16408** | GG | TT | TVu-12923 | GG | TT | GG |
| TVu-7991 | GG | TVu-4632 | GG | TT | muesseD | GG | TT | GG |
| **TVu-1477** | AA | TVu-7739 | GG | TT | TVu-7778 | GG | TT | GG |
| **TVu-10366** | AA | UCR_779 | GG | TT | **KVx-525** | AA | TT | GG |
| **KVx-525** | GG | TVu-13958 | GG | TT | TVu-7755 | GG | TT | GG |
| UCR_1432 | GG | TVu-3830 | GG | TT | TVu-1637 | GG | TT | AA |
| TVu-8656 | AA | **TVu-14691** | GG | GG | **INIA-40** | AA | TT | AA |
| **TVu-14691** | GG | TVu-6439 | AA | TT | **TVu-16408** | AA | TT | AA |
| TVu-6356 | AA | TVu-9393 | GG | TT | **TVu-1477** | AA | TT | GG |
| TVu-8612 | AA | TVu-3360 | GG | TT | **TVu-10366** | GG | TT | AA |
| TVu-8883 | AA | TVu-84 | GG | TT | **TVu-14321** | AA | TT | AA |
| TVu-10100 | AA | **TVu-14321** | GG | TT | IT95K-1093-5 | AA | TT | GG |
| **TVu-3965** | GG | TVu-1000 | AA | TT | **TVu-3965** | GG | TT | AA |
| TVu-2971 | AA | **TVu-1477** | GG | TT | **TVu-14971** | GG | TT | AA |

NN: number of nodules per plant, NE: percentage of nodule efficiency, NDW: Nodule dry weight in mg/plant. Bolded genotypes appear among the top 15 genotypes for more than 1 BNF-related trait.

the WRKY genes on Vu10_1598130, this finding suggests that the same gene may be present on different chromosomes within the cowpea genome, as previously observed in soybean [81]. The overlap of SNPs for NE and NDW suggests possible pleiotropy or closely linked QTLs. Hence, WRKY genes regulate stress-responsive gene expression through W-box binding, influencing plant growth, development, stress tolerance, and nodule formation essential for biological nitrogen fixation in legumes [82].

The second genomic position scanned by GWAS with a biological function related to NDW was Vu7_34437020. This position marked a gene named Vigun07g221300 located at 31.4 kb downstream away from the actual marker revealed by the GWAS scan. This gene's biological function was associated with a gene named SPX-domain, which plays a crucial role in nodule development and function by managing phosphate levels [83]. Previous works reported that this gene is located on multiple chromosomes in both soybean [83] and common bean [84]. The third genomic region identified through GWAS as associated with BNF was Vu11_28197064, marking the *Thoumatin* gene (*Vigun11g096700*), located 9.5 kb upstream at Vu11_28206512. *Thoumatin* genes have been reported to be involved in regulating rhizobial strain specificity, a crucial factor in symbiotic nitrogen fixation and plant development [68,69]. Comparative studies in cereals and legumes suggest that the chromosomal location of *Thoumatin* genes varies by species, yet their functional role in plant-microbe interactions remains conserved [85,86]. Our identification of Thoumatin on chromosome 11 in cowpea aligns with these previous findings reinforcing its importance in BNF. This underscores the relevance of SNP marker Vu11_28197064 in tracking genetic variations that influence nitrogen fixation efficiency in cowpea.

Knowledge of marker effect through segregation patterns is essential to convert them to Kompetitive Allele-Specific PCR (KASP) for genotyping the polymorphisms at various loci [87]. The marker effect of the identified genomic regions involves different allele variants. The marker Vu6_24933702 has AA allele contributing to increased NN per plant. The top five genotypes with that homozygous allele variant include INIA-40, TVu-14971, TVu-7991, TVu-1477 and TVu-10366. For an increased NE, the favorable alleles of marker Vu01_34265294 are AA or TT while the marker Vu10_1598130 has a favorable homozygous allele TT. The top five genotypes with an increased NE complying with these alleles include 524-B, Tvu-16408,

TVu-4632, TVu-7739 and UCR_779. In terms of NDW, the markers Vu7_34414059 and Vu11_28197064 have a homozygous status of GG allele with very few having AA corresponding to high NDW. In addition, the marker Vu7_34437020 with a homozygous allele of TT also contributed to high NDW. The top five genotypes with this favorable allele for NDW include 524-B, TVu-12923, MuesseD, TVu-7778 and KVx-525. Overall, from our results, TVu-1477 can be considered the most efficient genotype in nitrogen fixation as it combines favorable alleles for the three BNF indicators studied.

It is important to note that validation of the mapped loci and gene expression in an independent population with more accuracies in the screening in both greenhouse and field conditions, would increase the robustness and liability of these findings. Once validated in other cowpea populations, the identified significant SNPs showing a favorable homozygous allele can be exploited for developing Kompetitive Allele-Specific PCR (KASP) markers [88–91], useful for high genotyping efficiency [92] in the marker-assisted selection (MAS) process.

## Conclusion

This Genome-wide association mapping has identified 6 candidate genes associated with BNF traits in cowpea. Those genes include Vigun06g121800, Vigun01g160600, Vigun10g014400, Vigun07g221500, Vigun07g221300 and Vigun11g096700. Their different positions within and/or between chromosomes imply their status as complex traits. Confirming them in new cowpea populations or with an increased sample of the same mini-core cowpea population would help cowpea breeders get more insights to shorten the breeding scheme through marker-assisted selection strategies. The genotype TVu-1477 was identified to have favorable alleles for both three studied traits. Converting these markers into low-cost and friendly usable KASP markers would be the easiest way to select new high nitrogen-fixing genotypes having the targeted regions of interest.

## Supporting information

**S1 Table. List of the genotypes used in the GWAS study.**
(XLSX)

**S2 Table. SNP table_combined analysis.**
(XLSX)

**S3 Table. SNP table_experiment two.**
(XLSX)

**S4 Table. SNP table_experiment one.**
(XLSX)

**S5 Table. Cowpea gene_information.**
(XLS)

## Acknowledgements

We acknowledge the support of individuals from MaRCCI and the BNF lab at the Department of Agricultural Production/ Makerere University.

## Author contributions

**Conceptualization:** Gelase Nkurunziza, Emmanuel K Mbeyagala, Isaac Onziga Dramadri, John Baptist Tumuhairwe, Thomas Lapaka Odong.

**Data curation:** Gelase Nkurunziza.

**Formal analysis:** Gelase Nkurunziza, Emmanuel Amponsah Adjei.

**Funding acquisition:** Gelase Nkurunziza.

**Investigation:** Gelase Nkurunziza, Emmanuel K Mbeyagala.

**Methodology:** Gelase Nkurunziza, Emmanuel K Mbeyagala, Isaac Onziga Dramadri, John Baptist Tumuhairwe, Thomas Lapaka Odong.

**Resources:** Gelase Nkurunziza, Richard Edema, Arfang Badji, Rahiel Hagos Abraha.

**Software:** Gelase Nkurunziza, Emmanuel Amponsah Adjei, Isaac Onziga Dramadri.

**Supervision:** Emmanuel K Mbeyagala, John Baptist Tumuhairwe, Thomas Lapaka Odong.

**Validation:** Gelase Nkurunziza, Emmanuel K Mbeyagala, Isaac Onziga Dramadri, John Baptist Tumuhairwe, Thomas Lapaka Odong.

**Visualization:** Gelase Nkurunziza, Emmanuel Amponsah Adjei, Isaac Onziga Dramadri, Kpedetin Ariel Frejus Sodedji.

**Writing – original draft:** Gelase Nkurunziza.

**Writing – review & editing:** Gelase Nkurunziza, Emmanuel K Mbeyagala, Emmanuel Amponsah Adjei, Isaac Onziga Dramadri, Richard Edema, Arfang Badji, Rahiel Hagos Abraha, Astere Bararyenya, Kpedetin Ariel Frejus Sodedji, Phinehas Tukamuhabwa, Mildred Ochwo Ssemakula, John Baptist Tumuhairwe, Thomas Lapaka Odong.

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
