## [Decision Letter · Decision Letter 0]

23 Jan 2025

PONE-D-24-55003Genome-Wide Association Study of Biological Nitrogen Fixation-Related Traits in a Mini-core Subset of Cowpea GermplasmPLOS ONE

Dear Dr. Nkurunziza,

Thank you for submitting your manuscript to PLOS ONE. After careful consideration, we feel that it has merit but does not fully meet PLOS ONE’s publication criteria as it currently stands. Therefore, we invite you to submit a revised version of the manuscript that addresses the points raised during the review process. Please submit your revised manuscript by Mar 09 2025 11:59PM. If you will need more time than this to complete your revisions, please reply to this message or contact the journal office at plosone@plos.org . Please include the following items when submitting your revised manuscript:

We look forward to receiving your revised manuscript.

Kind regards,

Aditya Pratap

Academic Editor

PLOS ONE

2. We note that your Data Availability Statement is currently as follows: [All relevant data are within the manuscript and its Supporting Information files.] Please confirm at this time whether or not your submission contains all raw data required to replicate the results of your study. Authors must share the “minimal data set” for their submission. PLOS defines the minimal data set to consist of the data required to replicate all study findings reported in the article, as well as related metadata and methods (https://journals.plos.org/plosone/s/data-availability#loc-minimal-data-set-definition).

4. Please ensure that you refer to Figure 2 in your text as, if accepted, production will need this reference to link the reader to the figure.

Additional Editor Comments:

The manuscript is well written and presents important information. It may be accepted after minor revisions as suggested by both teh reviewers.

Reviewers' comments:

Reviewer's Responses to Questions

**Comments to the Author**

1. Is the manuscript technically sound, and do the data support the conclusions?

Reviewer #1: Yes

Reviewer #2: Yes

2. Has the statistical analysis been performed appropriately and rigorously? 

Reviewer #1: Yes

Reviewer #2: Yes

3. Have the authors made all data underlying the findings in their manuscript fully available?

Reviewer #1: Yes

Reviewer #2: Yes

4. Is the manuscript presented in an intelligible fashion and written in standard English?

Reviewer #1: Yes

Reviewer #2: Yes

5. Review Comments to the Author

Reviewer #1: Comments and Questions:

1. What measures were taken to standardize environmental conditions across the two experiments, and how might this have affected results.

2. Were any alternative heritability models considered, and how do these results compare?

3. Heritability was lower in the combined experiment compared to individual ones. Could this be due to genotype-by-environment interactions, and how were these accounted for?

4. How might the randomized complete block design (RCBD) have influenced heritability estimates? Were spatial or block effects significant, and how were they adjusted?

5. How do the heritability estimates influence the feasibility of marker-assisted selection (MAS) for improving BNF traits in cowpea?

6. How were the significance thresholds for identifying SNPs determined? Were methods such as Bonferroni correction or False Discovery Rate (FDR) applied consistently across all six GWAS methods?

7. How were population structure and kinship accounted for in the GWAS models? Were any metrics like genomic inflation factor (λ) calculated to assess potential biases due to population stratification?

8. Was the SNP density across the genome sufficient to capture the variation related to BNF traits? Were any important regions potentially missed due to low SNP coverage?

9. Several SNPs were associated with multiple traits (e.g., NN, NE, NDW). Could these represent pleiotropic effects, or are they indicative of linked but distinct QTLs?

10. Was genotype imputation used to fill in missing data for the SNPs? If so, how was the accuracy of imputation validated?

11. The Manhattan and QQ plots generated in the study—do they indicate any potential issues with model fit or inflation? Were there any unexplained deviations in the QQ plots?

12. How were the interactions between SNPs (epistasis) analyzed or accounted for? Could combining multi-locus effects yield more predictive markers?

13. The study reports moderate heritability for the BNF traits. How do these values compare to heritability estimates for similar traits in other legumes like soybean or common bean?

Reviewer #2: Comments for improvement

• Line Numbers 25 to 47, The abstract is comprehensive; however, it could be streamlined by emphasizing the key findings. For instance, one could minimize the discussion of methods and highlight significant results, such as the identified genomic regions. Emphasize the practical implications of your findings, particularly how these genomic insights can be directly utilized in cowpea breeding programs.

• Line numbers 62 to 79, The introduction discusses BNF efficiency; however, a clear comparison with earlier GWAS studies on legumes would enhance the rationale.

• Line numbers 133-134, Nodule formation in legumes is significantly affected by the rhizobial strain, as the compatibility between the legume plant and specific rhizobial strains directly influences nodule efficiency. What was the rationale for selecting a single rhizobial strain in your study, given the availability of multiple strains?

•Indicate if an independent dataset or validation population was utilized to verify the identified significant SNPs. Consider including this as a limitation if applicable.

•Present a justification for the utilization of the mrMLM.GUI package and elucidate the selection of six distinct methods.

• Line numbers 490 to 493, Elaborate on the functional importance of the identified candidate genes. How do these genes interact with other traits or processes that influence cowpea productivity?

•Several points are repeated in the discussion, including the function of WRKY genes and the necessity for KASP markers. Enhancing the flow and reducing redundancy can be achieved by streamlining these points.

•The references to supporting studies are significant; however, their integration into the narrative could be improved. Instead of enumerating genes identified in other species, directly compare their functions across species and their significance to cowpea. In the discussion, the association between the SNP markers and their corresponding genes should be articulated more clearly, avoiding redundancy in points related to various traits (e.g., WRKY and Thoumatin genes).

6. PLOS authors have the option to publish the peer review history of their article (what does this mean? ). If published, this will include your full peer review and any attached files.

**Do you want your identity to be public for this peer review?** For information about this choice, including consent withdrawal, please see our Privacy Policy .

Reviewer #1: No

Reviewer #2: No

---

## [Author Response · Author response to Decision Letter 1]

12 Mar 2025

The responses to reviewers one and two have been uploaded

---

## [Editor Report · Decision Letter 1]

18 Mar 2025

Genome-Wide Association Study of Biological Nitrogen Fixation  Traits in Mini-core Cowpea Germplasm

PONE-D-24-55003R1

Dear Dr. Nkurunziza,

We’re pleased to inform you that your manuscript has been judged scientifically suitable for publication and will be formally accepted for publication once it meets all outstanding technical requirements.

Kind regards,

Aditya Pratap

Academic Editor

PLOS ONE

Additional Editor Comments (optional):

All queries have been addressed and the manuscript can be accepted.

---

## [Editor Report · Acceptance letter]

PONE-D-24-55003R1

PLOS ONE

Dear Dr. Nkurunziza,

I'm pleased to inform you that your manuscript has been deemed suitable for publication in PLOS ONE. Congratulations! Your manuscript is now being handed over to our production team.

Kind regards,

on behalf of

Dr. Aditya Pratap

Academic Editor

PLOS ONE